# communications
## engineering

# Localization of seismic waves with submarine fiber optics using polarization-only measurements

Luis Costa [1✉], Siddharth Varughese[2], Pierre Mertz[2], Valey Kamalov[3] & Zhongwen Zhan[1]

Monitoring seismic activity on the ocean floor is a critical yet challenging task, largely due to the difficulties of physical deployment and maintenance of sensors in these remote areas. Optical fiber sensing techniques are well-suited for this task, given the presence of existing transoceanic telecommunication cables. However, current techniques capable of interrogating the entire length of transoceanic fibers are either incompatible with conventional telecommunication lasers or are limited in their ability to identify the position of the seismic wave. In this work, we propose and demonstrate a method to measure and localize seismic waves in transoceanic cables using only conventional polarization optics, by launching pulses of changing polarization. We demonstrate our technique by measuring and localizing seismic waves from a magnitude $M_w$ 6.0 earthquake (Guerrero, Mexico) using a submarine cable connecting Los Angeles, California and Valparaiso, Chile. Our approach introduces a cost-effective and practical solution that can potentially increase the density of geophysical measurements in hard-to-reach regions, improving disaster preparedness and response, with minimal additional demands on existing infrastructure.

---

[1] Seismological Laboratory, California Institute of Technology, Pasadena, CA 91125, USA. [2] Infinera Corporation, 9005 Junction Drive, Savage, MD 20701, USA. [3] Valey Kamalov LLC, Gainesville, FL, USA. ✉email: ldpcosta@gmail.com

The lack of geophysical instrumentation on the ocean floor imposes a bottleneck in our ability to study the Earth's structure and dynamics, impeding access to a wealth of geophysical and oceanographic information in remote off-shore regions. Overcoming this challenge would not only benefit fundamental research efforts, but also enable timely detection of off-shore earthquakes and tsunamis, providing adequate warning to nearby coastal areas.

Most technological developments for ocean floor vibration sensing are either temporary installations or are associated with high deployment and maintenance costs, which discourage their wide-scale use[1–3]. One of the emerging alternatives, however, proposes the use of optical signals propagating in existing telecommunication optical fibers for sensing, thus leveraging the immense scale of the telecommunications industry and infrastructure. By using in-land instruments and relying on the existing fiber cables (which cover an appreciable portion of the seafloor and reach locations of high geophysical interest), these methods aim to reduce deployment costs, ease maintenance, and enable continuous monitoring of the seafloor.

Indeed, optical fiber strain sensing technologies have proven successful in geophysical measurements for both in-land and near-shore applications[4–8]. Long-haul transoceanic cables, however, introduce unique challenges that cannot be tackled using the prevailing techniques applied for in-land cables (i.e., distributed optical fiber sensing[5,9]). These methods often rely on weakly backscattered light, which restricts their application to the first ~100 km of cable, makes them incompatible with optical repeaters[10], and elicits the use of high optical peak powers (leading to optical nonlinearities which compromise coexisting data channels in the same fiber strand[11]). These limitations are often inconsequential for in-land or near-shore deployments, as the total cable lengths are shorter and the abundance of unused fiber strands facilitates the use of dedicated sensing fibers, but constitute critical roadblocks for transoceanic deployments. Moreover, successfully overcoming the instrumentation gap using transoceanic optical fibers demands an emphasis on scalability and compatibility with the existing infrastructure, instead of the use of specialized instruments and sources.

Recent demonstrations of optical fiber sensing in subsea network cables can be classified as either interferometric[12, 13] or polarization-based approaches[14, 15]. Earlier demonstrations used the full span of the cable as a single sensor (unable to localize the point of perturbation), by employing either ultrastable lasers for long-haul interferometry[12] or observing the change of the state of polarization (SOP) at the output of the fiber[14, 15]. Full-span methods, however, have multiple disadvantages: without localization capabilities, multiple cables or sensors must be used for epicenter localization, which is further complicated by the long gauge length. Furthermore, by measuring the cumulative effects of environmental noise sources over the whole cable length against the localized perturbations of interest, these methods will necessarily suffer from higher noise floors.

In a recent demonstration of localization, interferometric measurements were able to pinpoint measurements to a span of cable between consecutive repeaters (typically spaced every 50 to 100 km) by measuring the return light reflected from high-loss loop back (HLLB) paths placed at each repeater[13]. Nonetheless, this approach still required specialized laser sources of much greater coherence length (and cost) than conventional telecommunication transponders, restricting its wide deployment. On the other hand, polarization-based methods benefit from less stringent hardware requirements but have thus far been unable to localize the seismic wave to a single-span. The non-commutative nature of birefringence operations has limited these methods to

full-cable measurements or, at most, to the localization of a single dominant perturbation occurring along the cable.

In this work, we propose and demonstrate a method that relies only on a series of SOP measurements to localize geophysical measurements to within a single span of an amplified transoceanic fiber cable, with minimal alterations to conventional optical transponders[16]. We detect, measure and localize a magnitude $M_w$ 6.0 earthquake that happened in Guerrero, Mexico, on the 11th of December 2022. The earthquake was detected on a transoceanic fiber cable that connects Los Angeles (California) to Valparaiso (Chile). The proposed method (hereby dubbed the eigenvalue method) demands only incoherent polarization measurements and can coexist with co-propagating data channels in the same fiber strand.

## Results

**Single-span localization**. Our method employs a laser emitting light pulses of fixed polarization, which are then passed through a programmable polarization controller and launched into the fiber under test (FUT). At the receiver end, a polarimeter is used to measure the state of polarization (SOP) of the $N$ reflections that originate from each HLLB paths present at every repeater in the majority of modern submarine cables (as shown in Fig. 1a).

We express the SOP of the reflection originating from the $m$th repeater, $\hat{y}_{(m)}$, as a normalized Stokes vector given by:

$$\hat{y}_{(m)} = \mathbf{A}_{(m)}\hat{s}, \tag{1}$$

where $\hat{s}$ is the normalized Stokes vector representing the SOP of the input pulse, and $\mathbf{A}_{(m)}$ is the real-valued rotation (orthogonal) matrix that describes the cumulative birefringence effects of the complete round-trip to and from the $m$th repeater (Fig. 1a).

The matrices $\mathbf{A}_{(m)}$ can be measured by probing the fiber with multiple inputs $\hat{s}_i$ spanning the full Stokes space, instead of a single input SOP. This is optimally achieved by cycling the input SOP between three states forming an orthogonal Stokes basis, $\hat{s}_i$ where $i = \{1, 2, 3\}$ (e.g., horizontal, vertical, and left-circularly polarized light), assuming that the fiber remains stationary over the three acquisitions, but can be achieved with any choice of 3 vectors that spans the full Stokes space.

Each matrix $\mathbf{A}_{(m)}$ can be decomposed into the birefringence contribution from the laser to the $m$th repeater $\mathbf{A}_{(m)}^{\mathrm{fwd}}$ (forward path) and the birefringence contribution from the $m$th repeater to the receiver through the HLLB path $\mathbf{A}_{(m)}^{\mathrm{bkd}}$ (backward path):

$$\mathbf{A}_{(m)} = \mathbf{A}_{(m)}^{\mathrm{bkd}}\mathbf{A}_{(m)}^{\mathrm{fwd}}. \tag{2}$$

If we now define the matrices encoding only the local birefringence of the fiber span between repeater $(m-1)$ and $(m)$ as $\mathbf{X}_{(m)}^{\mathrm{fwd}}$ and $\mathbf{X}_{(m)}^{\mathrm{bkd}}$ (for the forward propagation and backward propagation, respectively), we can rewrite Eq. (2) as:

$$\mathbf{A}_{(m)} = \mathbf{A}_{(m-1)}^{\mathrm{bkd}}\mathbf{X}_{(m)}^{\mathrm{bkd}}\mathbf{X}_{(m)}^{\mathrm{fwd}}\mathbf{A}_{(m-1)}^{\mathrm{fwd}}. \tag{3}$$

Measuring the local birefringence matrix $\mathbf{X}_{(m)}^{\mathrm{fwd}}\mathbf{X}_{(m)}^{\mathrm{bkd}}$ is not possible, since the previously described strategy of probing the fiber with pulse triplets allows us only to measure the cumulative $\mathbf{A}_{(m-1)}^{-1}$ and $\mathbf{A}_{(m)}$. However, it is possible to retrieve partial information about $\mathbf{X}_{(m)}^{\mathrm{fwd}}\mathbf{X}_{(m)}^{\mathrm{bkd}}$ from the measured $\mathbf{A}_{(m)}$ matrices by performing the following operation:

$$\mathbf{A}_{(m-1)}^{-1}\mathbf{A}_{(m)} = \mathbf{U}^{-1}\mathbf{X}_{(m)}^{\mathrm{bkd}}\mathbf{X}_{(m)}^{\mathrm{fwd}}\mathbf{U}, \tag{4}$$

where $\mathbf{U} = \mathbf{A}_{(m-1)}^{\mathrm{fwd}}$ can be any (unknown) unitary matrix. The resulting matrix from Eq. (4) is similar to $\mathbf{X}_{(m)}^{\mathrm{fwd}}\mathbf{X}_{(m)}^{\mathrm{bkd}}$ and therefore

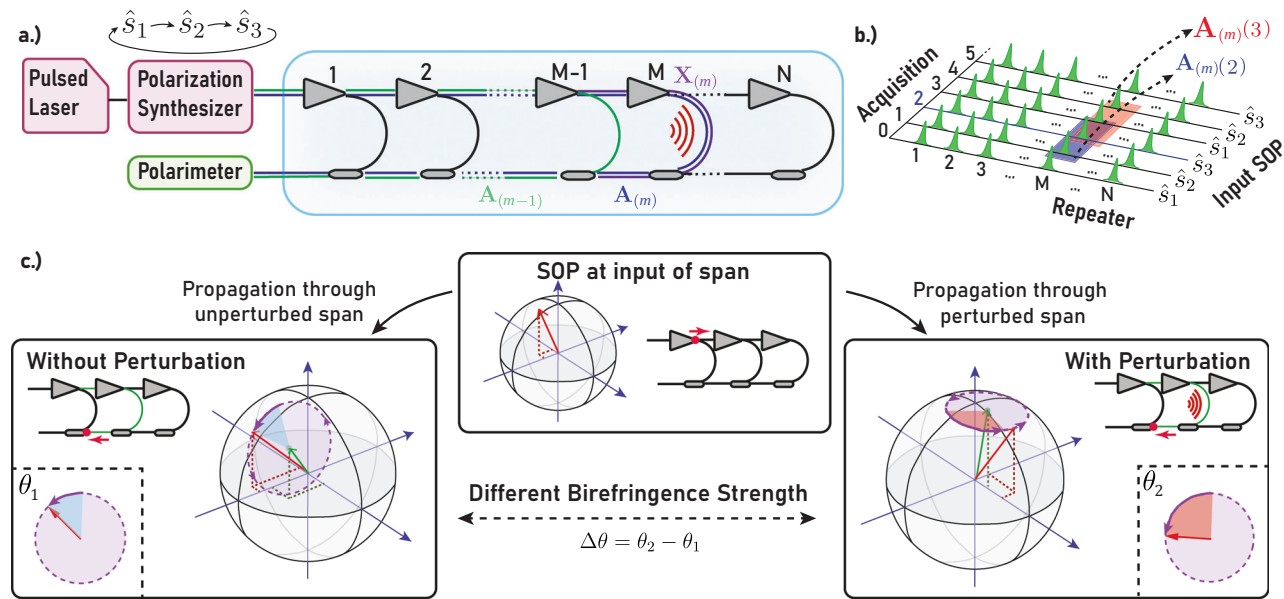

**Fig. 1 Measurement concept. a** Representation of the sections of fiber modeled by each of the matrices. The $\mathbf{A}_{(m)}$ matrix models the blue colored fiber path, the $\mathbf{A}_{(m-1)}$ matrix models the green fiber path, and the local $\mathbf{X}_{(m)}$ matrix models the purple fiber path. **b** Illustration of the acquisition of an $\mathbf{A}_{(m)}$ matrix, from three consecutive acquisitions. Each acquisition recovers $N$ reflections, each originating from a specific repeater in the cable. The $\mathbf{A}_{(m)}$ matrix is constructed by combining the Stokes vectors from the 3 consecutive reflections originating from the same ($m$th) repeater in a matrix, when probing the fiber with different SOP. **c** Conceptual visualization of the information obtained about the local birefringence using our proposed approach. Propagation through a span of fiber is modeled as a rotation of the Stokes vector representing the SOP at the span input (red arrow) around the effective birefringence vector of the span (green arrow), tracing the path in represented in the purple circles. A perturbation acting on the fiber affects the orientation and strength of the birefringence vector (bottom, left and right). The change in angle of the rotation around the birefringence vector, which we measure, is proportional to a change in the birefringence strength.

has the same eigenvalues:

$$\text{eig}\left\{\mathbf{X}_{(m)}^{\text{fwd}}\mathbf{X}_{(m)}^{\text{bkd}}\right\} = \text{eig}\left\{\mathbf{A}_{(m-1)}^{-1}\mathbf{A}_{(m)}\right\}. \quad (5)$$

The two complex eigenvalues represent the local angle of rotation of polarization around the unknown birefringence vector (see Fig. 1c), and are thus proportional to the birefringence strength along a single, localized span. A change in the local birefringence that affects the overall magnitude of the rotation of SOP in the Stokes space can be measured by computing the angle of the eigenvalue at time $t$ for every $m$th reflection, and comparing it to a previous reference measurement at time $t_{\text{ref}}$. Hence, a measurement $\mu(m, t)$ can be defined as:

$$\mu(m, t) = \measuredangle\left(\text{eig}\left\{\mathbf{A}_{(m-1)}^{-1}\mathbf{A}_{(m)}\right\}(t)\right) - \measuredangle\left(\text{eig}\left\{\mathbf{A}_{(m-1)}^{-1}\mathbf{A}_{(m)}\right\}(t_{\text{ref}})\right) \quad (6)$$

Figure 1 visually depicts the information encoded in each of the $\mathbf{A}_{(m)}$ and $\mathbf{X}_{(m)}$ involved in the calculation, the method for recovering the $\mathbf{A}_{(m)}$ matrix, and a depiction of the measurement concept.

**Measurement and post-processing**. After measuring the SOP of 3 consecutive probe pulses of orthogonal SOP (Fig. 1b), the obtained $A_{(m)}(t)$ matrices are generated by stacking the normalized stokes vectors as:

$$A_{(m)}(t) = \begin{bmatrix} \vdots & \vdots & \vdots \\ \hat{y}_{(m)}(t-1) & \hat{y}_{(m)}(t) & \hat{y}_{(m)}(t+1) \\ \vdots & \vdots & \vdots \end{bmatrix}. \quad (7)$$

Before processing, each of the recovered $A_{(m)}(t)$ matrices is denoised by finding the closest unitary matrix, in the Frobenius norm sense. This is achieved by performing the singular value

decomposition of the $A_{(m)}(t)$ matrix ($A_{(m)}(t) = U_m \Sigma_m V_m^{\text{T}}$), and keeping the denoised matrix $\hat{A}_{(m)}(t) = U_m V_m^{\text{T}}$.

We then perform the operation described in Eq. (4) for each repeater using the denoised matrices. We store the angle of the complex eigenvalue of the positive argument, and calculate the difference in the measured angle to that of the first acquisition. The complete processing stack is illustrated in the Supplementary Information, Supplementary Note 1.

**Experimental results**. On December 11, 2022, at 14:31:29 UTC, a magnitude 6.0 earthquake occurred in Guerrero, Mexico, which we captured on the Curie transoceanic fiber cable, which connects Los Angeles (California) to Valparaiso (Chile) (Fig. 2b). The cable contains 110 repeaters with HLLB paths, each comprising a high-splitting ratio directional coupler and a Bragg grating reflector. The interrogation setup (situated in the Los Angeles terminal) is depicted in Fig. 2a, and includes a telecommunication transponder[17] used to send linearly polarized optical pulses through a polarization synthesizer on the emitter side, and a polarimeter on the receiver side, which is used to evaluate the state of polarization of the received reflections.

By observing the changes in the SOP of the reflections coming from the HLLB paths when using constant input polarization (hereby called direct SOP measurements), the earthquake was visible on span 41 and all following spans (due to the cumulative nature of this method). We observe an initial arrival about 153 s after the earthquake origin time. Given that the 41st span is about 532 km away from the earthquake epicenter (Fig. 3a), this arrival time is consistent with a dominant seismic phase Sg wave in oceanic crust. The observed earthquake is predominantly visible in the 0.25–0.35 Hz band, and its duration is about 10 min (Fig. 3b). This is consistent

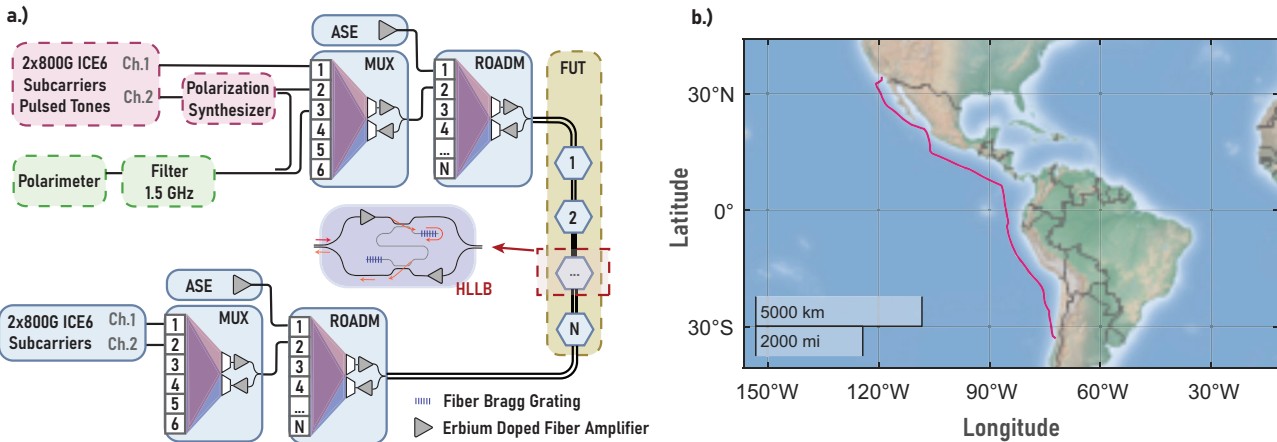

**Fig. 2 Experimental setup and layout of the interrogated fiber cable. a** Diagram of the experimental setup. The emitter part of our interrogation system is color-coded red, and the detector part of our interrogator system is color-coded green. The fiber under test (FUT) is color-coded yellow. All the relevant boxes use a dashed outline. The blue color-coded boxes represent parts of the telecommunication link that are not fundamental parts of the proposed measurement technique (ASE amplified spontaneous emission, MUX multiplexer, ROADM reconfigurable optical add-drop multiplexer). **b** Map of the FUT (map generated using MATLAB's *geoplot* function).

with previous reports of direct SOP measurements performed over the full-span of the cable for similar events[14].

We tested the localization abilities of our proposed eigenvalue method by plotting the signal power in the earthquake band (0.25 to 0.35 Hz) over 60-s-long time windows, depicted in Fig. 3c, left. The earthquake signal is first visible in the 41st span, in accordance with the direct SOP measurements (Fig. 3c, right). However, while the earthquake signal is visible in every span following the 41st when using the direct SOP approach, our eigenvalue method localizes the measurement to a single span with minimal crosstalk (defined as the increase in signal noise power in the earthquake frequency band to subsequent fiber locations). We observe a median value of ~1 dB of crosstalk (Fig. 3d).

We also used the eigenvalue method to observe the seismic wave move out. Besides span 41, we can observe the earthquake signal in span 43 (673 km away from the epicenter), starting ~40 s after the wave recorded in span 41 (532 km away from the epicenter), as shown in Fig. 3e. Notably, however, most neighboring spans to the 41st do not display a clear earthquake signal, likely due to a combination of the earthquake radiation pattern of S waves, the relative orientation of the fiber, variations in mechanical coupling to the seafloor, and the complicated response of birefringence to different types of stimulus.

## Discussion

In this work, we proposed and demonstrated a scalable approach to sense and localize perturbations affecting the birefringence of amplified telecommunication fiber cables. Our method is compatible with telecommunication-grade sources and detectors, requiring only the addition of a polarization synthesizer to conventional transponders. With about 500 operational cables worldwide, mostly with HLLB reflectors, each of 50–100 spans, our measurements can be extended to 25,000–50,000 fiber spans without touching wetplant infrastructure.

We successfully accomplished the localization of a seismic wave, identifying its location to a single span of fiber between two optical repeaters. The ability to localize the seismic wave to within a span enables the observation of the seismic wave move-out (as demonstrated in Fig. 3e), and may lead to further benefits, such as the determination of the epicenter of a seismic event using a single fiber, and reduced influence of environmental noise compared to cumulative approaches. Our approach displayed minimal crosstalk to subsequent spans as a ~1 dB increase in noise-

floor in the earthquake band. The physical origin of the crosstalk is likely linked to the non-stationarity of the fiber over the pulse triplet required to make a measurement (as demonstrated numerically in the Supplementary Information, Supplementary Note 4), suggesting that it may be improved by increasing the rate of measurement pulses (limited by the polarization synthesizer in our implementation).

Regarding the detection limit of the two tested approaches, we observed similar SNR between the eigenvalue method and direct SOP measurements. Notably, both approaches failed to detect the earthquake at spans closer to the epicenter than the 41st. This suggests that the dominant contribution to the detection limit in any SOP-based method could originate from the complex sensitivity of the local birefringence to different environmental stimuli (e.g., bends[18], twists[19], or other effects[20]), the non-linear nature of SOP-based measurements, variations in mechanical coupling along the cable, or the geometry/layout of the cable with respect to the induced deformation by the seismic wave[21].

Nonetheless, it is not easy to draw a direct comparison between the detection limit of both approaches, given the fundamental differences between the eigenvalue and direct-SOP methods: on the one hand, the detection limit when using direct-SOP methods with HLLB will likely be in part determined by the accumulated length of cable up to the interrogated span and the environmental noise acting on the cable (due to the cumulative nature of environmental noise). On the other hand, the eigenvalue method's insensitivity to changes to the birefringence vector orientation suggests potentially lower sensitivity in some scenarios, where the net effect along the span predominantly rotates the birefringence vector, without a great net effect on birefringence strength.

A possible future research direction would be to focus on analyzing lower frequency events. Spectral features in the mHz range are of particular interest for measurement of tsunami and infragravity waves. Per-span resolution of tsunami wave propagation measurement in the open sea using polarization measurements has some advantages when compared to phase-based approaches[13], by being immune to laser phase noise due to measuring the relative dynamics of two electric field vectors.

Additionally, the sensitivity of the eigenvalue technique may potentially be improved by using a Jones receiver or by recovering the full Muller matrix. Currently, by acquiring three sets of Stokes

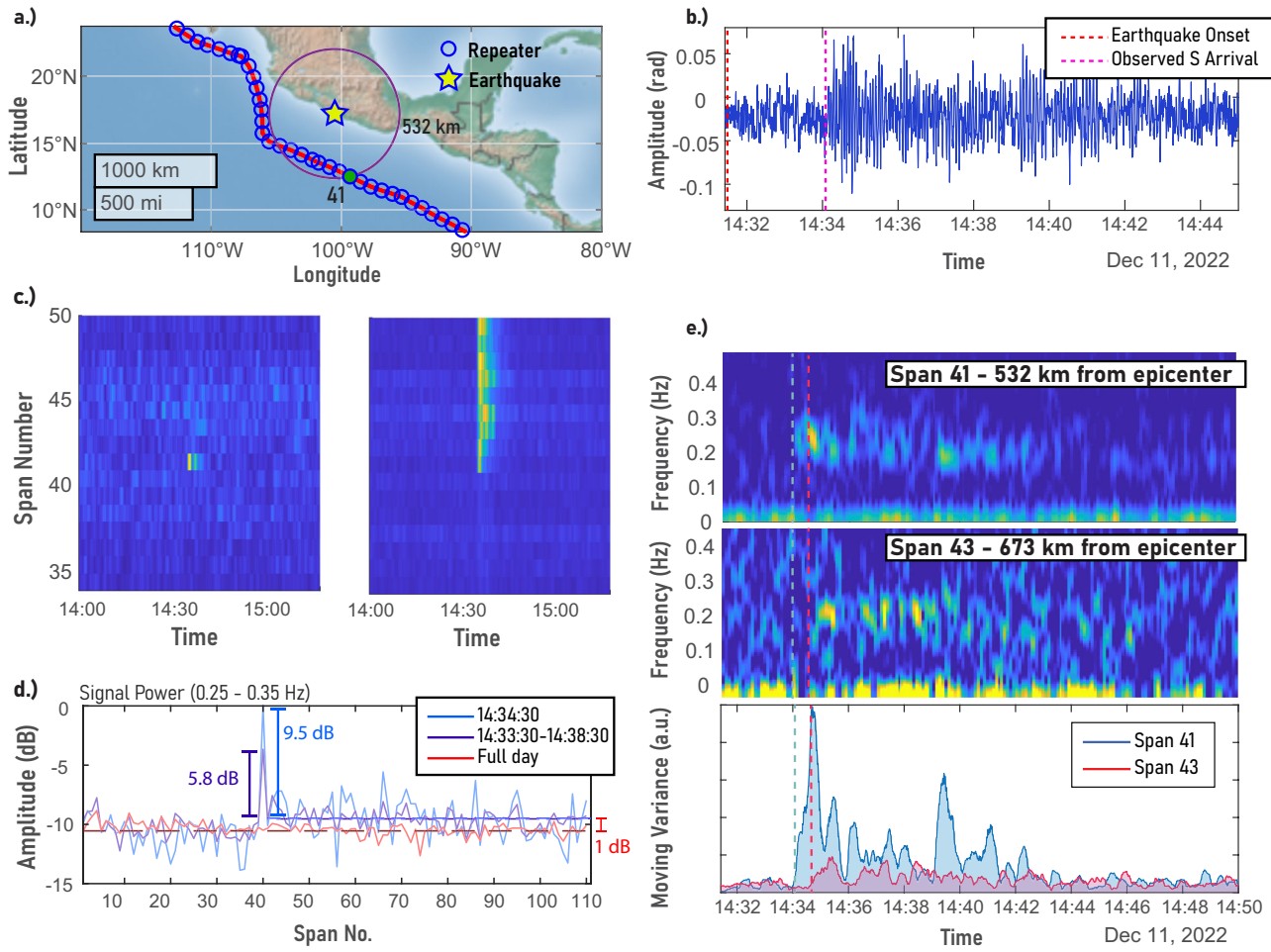

**Fig. 3 Experimental results. a** Close-up map of the earthquake and affected cable span (map generated using MATLAB's *geoplot* function). Purple circle shows the distance between the 41st span and the epicenter. **b** Time-series of the wave measured in the 41st span, using the eigenvalue method. The red line marks the onset time of the earthquake, and the pink line marks the apparent onset time of the recorded wave. **c** 2D plots of the signal power in the 0.25 to 0.35 Hz band, using (left) the eigenvalue method and (right) direct measurements of the SOP values, over 60-s-long time windows. Using direct SOP measurements, all spans following the affected span register the perturbation, while the eigenvalue method localizes the perturbation to a single span. **d** Measurement of crosstalk using the eigenvalue method, normalized to the maximum signal variance measured in span 41 over the earthquake band. We observe a 1 dB average increase in noise floor to the following spans during the event. **e** Spectrograms of span 41 (Top) and span 43 (Middle), and observation of seismic wave move out. (Bottom) Moving variance of the signal above 0.1 Hz, calculated with a 20-s moving window. Dashed lines denote the apparent start of the earthquake signal in each span.

components, normalizing each of them, and calculating the closest unitary matrix, we are making an assumption of no polarization dependent loss in each span. The polarization rotation originating from environmental changes (i.e., changes to the birefringence) will be combined with the apparent rotation originating from polarization-dependent loss of the lumped elements as the same signal (which is largely time-independent, and not directly correlated to environmental changes).

In summary, we successfully demonstrated a minimally intrusive method to convert the existing telecommunications infrastructure into sensor arrays for geophysical measurements. Our present results demonstrate the ability to localize seismic events within fiber cables, overcoming known challenges of previously demonstrated SOP-based techniques, and clear paths for future optimization. Denser and more frequent seismic measurements of the oceanfloor can provide valuable insights into geological activity and the interior composition of the Earth, and enable cost-effective early warning systems to prevent tsunami and earthquake damage to coastal communities.

## Methods

**Choice of pulse width and repetition rate**. The pulse width and repetition rate were selected to accommodate the length and repeater spacing of the FUT. The signals from all repeater (HLLB) return paths are transmitted through the same fiber and are discriminated by their time-of-arrival since the respective input pulse launch (see Fig. 1b). In order to ensure that all reflections arrive before launching the next pulse into the FUT, the repetition rate must be selected as:

$$f_{\mathrm{rep}} \leq \frac{c_0}{2 n_{\mathrm{g}} L_{\mathrm{FUT}}}, \tag{8}$$

where $L_{\mathrm{FUT}}$ is the total length of the FUT, $n_{\mathrm{g}}$ is the group index of the fiber, and $c_0$ is the speed of light. By a similar argument, the maximum pulse width must ensure that reflections originating from consecutive repeaters do not overlap at the receiver. This is achieved by setting the pulse width $\tau_{\mathrm{p}}$ according to:

$$\tau_{\mathrm{p}} \leq \frac{2n}{c_0} \Delta L_{\mathrm{min}}, \tag{9}$$

where $\Delta L_{min}$ is the minimum distance between any pair of repeaters in the cable. According to these requirements, we set the pulses to be 300 µs long, and repeated at 9.52 Hz (105 ms).

**Acquisition rate**. Due to the 200 ms wait time between commands required by our polarization synthesizer, we could not synthesize the SOP of each individual pulse launched into the fiber. Instead, we synthesize the polarization of every second or third pulse launched into the fiber, and record only the reflections originating from pulses launched with the correct SOP.

This limitation translates to an effective probe pulse rate of 2.907 Hz (i.e., the average rate at which we launch a pulse with the intended polarization) and a measurement bandwidth of 0.97 Hz (at which we are able to acquire a full set of the required triplet without any repeating vectors). Our acquisition of the $A_{(m)}(t)$ matrices, however, is done at the oversampled rate of 2.907 Hz, by processing them in a moving window approach as shown in Fig. 1b. A step-by-step explanation of the acquisition process is provided in Supplementary Note 2.

**Direct SOP post-processing**. The direct SOP processing method requires the input SOP to remain constant across measurements. In our case, we separate the three possible input probe SOP and process each corresponding dataset independently. Each probe SOP generates three time-series, one for each of the three output Stokes components. The resulting nine time-series are processed independently.

The 2D plot displayed in Fig. 3c (right) is an average of the nine 2D plots obtained from processing each time series.

**Input basis set**. The polarization synthesizer was programmed to cycle through a set of three SOP that form a Stokes basis. A potential quality metric for the input vector set is defined as:

$$Q = |\det\{A_0\}|, \tag{10}$$

where $A_0$ is a matrix of the three column vectors $\hat{s}_i$ launched into the fiber. Using this metric, $Q = 1$ indicates a perfectly orthogonal basis set and $Q = 0$ indicates a set of vectors that do not span the full Stokes space. In our measurements, the input $Q$ was consistently kept above 0.9 (see Supplementary Note 3). Maximizing the value of $Q$ enhances the robustness of the measurement against optical noise and helps minimize crosstalk, as demonstrated through numerical simulations in the Supplementary Information, Supplementary Note 4.

## Data availability

The data that support the findings of this study are available from the corresponding author upon reasonable request.

## Code availability

The code that supports the findings of this study is available from the corresponding author upon reasonable request.

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

## Acknowledgements

Z.Z. acknowledges the support of the NSF CAREER award and the Moore Foundation.

## Author contributions

L.C. designed the algorithm and processed the data. L.C. prepared the manuscript with support from all authors. S.V. and P.M. set up and conducted the experiments. Z.Z., V.K. and P.M. led and supervised the project. All authors contributed to the experiment design. All authors participated in the data analysis and interpretation.

## Competing interests

S.V. and P.M. are employed by Infinera Corporation. V.K. is employed at Valey Kamalov, LLC.
