## [Peer Review File · Communications Engineering]

Reviewers' comments:

Reviewer #1 (Remarks to the Author):

I am very excited for this work, and I genuinely believe this technology and algorithm proposed will be crucial in leveraging long range telecommunications fibers for seismology. The results are timely, and although the concept of leveraging loop-back points builds off another cited recent work (Marra, Science 2022), application here to state-of-polarization sensors will be crucial moving forwards. The possibility of such instruments being minimally intrusive and affordable means this may be the path forward for large scale application.

The article is generally well written and I see very few problems in the way of grammar or explanations. Importantly, however, I feel strongly the authors need to be careful in their wording about having located an earthquake: identifying a fiber span where the signals are strongest is absolutely not equivalent to "precise localization." It is a great leap forward, granted, but for any seismologist reader this is vastly different from a useful earthquake location; even the title of the article could be seen as questionable in this regard.

Other specific questions:

(These are points I offer to potentially improve the manuscript; none do I feel as strongly about and should not impede publication)

Relating to my point about claiming to have located the earthquake, I appreciate the authors' transparency and honesty at line 118, about how some closer spans failed to see the signals. Also, it is clear from the map view (Fig 3a) that Span 41 is not the closest. Nevertheless, this is further evidence the authors should be careful in claims and wording.

The authors list some possible reasons for why other spans may not see the earthquake, additionally potentially there could be something relating to fiber curvature or geometry as mentioned in Fichtner et al (2022, "Introduction to phase transmission..."). Notably, that straight segments should see nothing, though I admit I don't know if this is applicable also for SOP-based measurements, nor how loop-back repeaters would affect this claim.

Line 42: Grammar: "On the other hand, polarization-based methods with less stringent hardware requirements have previously been unable of single-span localization" -> unable to [leverage / achieve / take advantage of] single-span approaches?

Line 42: Conceptual: You claim that polarization-based methods have been unusable for single-span localization until now, thus motivating the need for the eigenvalue method you propose. But Fig 3C-right, if I understand correctly, is using more traditional direct SOP methods, so what is the limitation there? I guess you cannot isolate Span 41 from that figure? Otherwise what is needed or gained by the

eigenvalue approach?

Figure 2: Comment: Some of the symbols and acronyms in the figure are unfamiliar to me as a seismologist. For example, I don't understand what is going on inside the HLLB (loopback inset). This is OK and potentially normally readers of Communication Engineering will be more used to the symbols and acronyms, I only mention it for perspective.

Line 85: When describing the instrumentation: are signals at each HLLB reflector sent back along the same single fiber, or does each get a unique fiber? I guess it's the former, but as someone less familiar with the technologies involved I can't see how you separate the different HLLB paths.

Fig 3e (and other analyses): The frequency range shown to have high energy at 0.25 to 0.35 Hz is rather narrower than I would expect for a M6.0 earthquake. Are there limitations on the instrument noise that limit these observations, or maybe some aspect of the eigenvalue approach? A spectrum or seismogram from a nearby land station (or one of comparable epicentral distance) would help convince me of the measurement capability.

Line 123: Clarification: "The eigenvalue method's insensitivity to changes..." -> does this imply the eigenvalue method would further outperform SOP approaches, or the other way around? What is meant by "specific stimuli"? If the eigenvalue approach is less sensitive but the direct SOP is more noisy, what will win? I sincerely appreciate this type of analysis and discussion, just I want to make sure I (and readers) understand the implications.

Related to the comment above: It would be great if the authors can comment on the sensitivity of such measurements as relating to earthquakes. Detecting a M6.0 is great for proof-of-concept, but such undersea fibers will only really add seismological value if they can detect things below the range of current traditional instruments (e.g. M1, M2, M3, etc. offshore). I realize this may be beyond the scope of this initial study, so this is not required, I only mention it would be interesting to comment on.

Line 166: Again as one less familiar with the technologies, I don't understand why 9 SOP time-series are generated and why averaging them for Fig 3C-right is appropriate. But possibly this is given in background / cited literature?

Reviewer #2 (Remarks to the Author):

The paper presents an interesting, telecom-compatible method for localizing geophysical disturbances across a potentially trans-continental fiber cable. The method is an adaptation of what had been already demonstrated in Ref 13 (using the loop-back channel in amplified submarine links to localize disturbances) except that in this case, the measurement is done using polarization changes and conventional telecom lasers (i.e. no need of high-coherence lasers as in Marra's paper). This implies some advantages as telecom lasers themselves can be used (however, the use of dedicated polarization synthesizers and polarimeters in the measurement channel is needed, which means some unusual

hardware in these nodes). In essence the authors use long pulses with a selectable polarization state (pulses are the size of the span length) and the reflection from each loop-back channel is analyzed as a function of the time of flight of these long pulses. I think the paper is interesting and has to be published. I have several concerns and questions that the authors can surely address in a relatively easy way:

- I had to read too much through the methods section to actually understand the measurement procedure, I am not sure if some of this information could be shifted to the Results section considering that people read Results before Methods. In this case, for people with some skill in optical measurements, the information in Methods is of key importance to understand the process. I suggest to move some of the hardware operation to Results and possibly move part of the matrix treatment to the Methods section.
- When it comes to localization, obviously the golden standard in all these systems is using a DAS. Overall, the pulses used here are 300 microseconds long, which is comparatively very long for a DAS (3 orders of magnitude larger). I wonder if a DAS-like architecture with such a relatively long pulse could also give a measurable signal. DAS would have the advantage of being more quantitative and linear than this scheme. I think that an evaluation of the backscattered energy in such a case could help the authors decide if a poor resolution DAS could also do the same measurement (of course with a more expensive laser).
- I am sure that during the measurement campaign there were other disturbances of smaller magnitude that could be recorded along the used cable (this is a seismically very active region). Please provide information of what is the minimum magnitude of event that could be detected in the measurement campaign done here. Showing the magnitude 6 event is interesting, but giving the actual sensitivity threshold would be necessary to comparatively assess this method and the others published in the literature.
- Sampling is very low (sub Hz in this case, potentially 2-3 Hz if the hardware had no delay times) as the reflections from all the repeater spans have to be collected and 3 polarization states have to be swept. Please comment if there is any room for increasing the sampling while keeping the same constraints in terms of fiber size.
- Of course the interest of gathering measurements across many points is using array methods. However, considering the "nonlinear" nature of these polarization measurements, would this be compatible with array processing?

Reviewer #3 (Remarks to the Author):

Review of "Localization of Seismic Waves in Submarine Fiber Optics Using Polarization-only Measurements," by Luis Costa et al.

The manuscript presents a report on the detection and localization of an earthquake using an undersea fiber optics infrastructure. It appears to be a valuable addition to the rapidly growing literature on this topic. I am inclined to endorse its publication pending the proper addressing of my concerns outlined

below:

1) My primary concern relates to the authors' use of singular value decomposition (SVD) as an intermediate step to obtain a polar decomposition for extracting the unitary part of the transmission matrix. While this procedure is standard in Jones space, it may not be suitable in Stokes space. To illustrate this, consider the simple case of combining a (partial) polarizer represented in Jones space by a (positive definite) matrix A and a concatenation of waveplates represented by an arbitrary unitary matrix U .

In Jones space, the transmission matrix T is given by $T = UA$. The polar decomposition of T is either $T = UA$ (right polar decomposition) or $T = BU$ (left polar decomposition) and is unique. Consequently, U is also unique, and applying SVD would yield the exact result, providing the unitary matrix U and, in Stokes space, the rotation matrix corresponding to U .

However, if the SVD is directly applied in Stokes space, it would not return the 3 by 3 rotation matrix corresponding to U . This limitation arises because representing a pure polarizer as a linear operator is not possible within the 3-dimensional Stokes space. To maintain the linearity of the representation, it becomes necessary to extend the Stokes space with an extra dimension representing the total power and replace the matrices that represent rotations in Stokes space with 4 by 4 Mueller matrices. In the extended space, the unitary component of the decomposition is the direct sum of a rotation in the 3-dimensional Stokes space and an identity in the fourth coordinate. This makes not straightforward the application of the SVD to extract from the transmission matrix the unitary part of the concatenation.

Of course, the use of the SVD in the 3-dimensional Stokes space would still produce, for the concatenation polarizer-waveplates, a unitary matrix, but in most cases this unitary matrix includes the polarization rotation induced by the partial polarizer, which would instead be filtered out if the SVD is applied in Jones space.

Earthquakes primarily affect fiber propagation by inducing changes in the fiber's refractive index and birefringence, thereby impacting the unitary part of the transmission matrix. On the other hand, polarization-dependent loss is mainly caused by lumped devices and remains substantially time-independent. By applying the SVD in Stokes space, crosstalk is generated between the time-independent polarization-dependent loss and the time-dependent unitary part of polarization rotation. Consequently, this crosstalk has the potential to significantly reduce the sensitivity to time-dependent birefringence changes.

Given that it is not challenging to extract the transmission matrix in Jones space from the data, the authors should reconsider their data processing approach and extract the unitary part of the fiber propagation by applying the SVD in Jones space rather than in Stokes space.

2) The experiment's specific details regarding the system where it was performed have not been provided in the report. However, it appears that the system under test bears a striking resemblance to Curie, the system described in [14]. The only discernible difference is the location of one of the system's terminals, with one being in Santiago instead of Valparaiso. To ensure transparency and enable readers

to thoroughly understand the characteristics of the system under test, it is crucial to provide this information. Additionally, the report should explicitly state whether the data were collected from the Santiago or Los Angeles terminal.

Minor comments:

Line 70: (disregard if the paper is modified following the suggestion in comment 1).

The authors' analysis is conducted in Stokes space, not in Jones space. Consequently, U represents an arbitrary matrix with real entries describing a (proper) rotation in Stokes space, which is a special case of an orthogonal matrix, not an arbitrary (complex) unitary matrix. This distinction is important as it ensures that readers are given the immediate perception that the analysis takes place in Stokes space, not in Jones space.

Line 112:

Would be beneficial that the definition of crosstalk is given the first time it is introduced and discussed. The reader is not exposed to the mathematical definition of crosstalk until hitting the figure caption of Fig. 4 of the supplementary material.

Line 157: (disregard if the paper is modified following the suggestion in comment 1)

Again, the U and V matrix are real, so that V^* should be the transpose of V . Since the star is usually reserved for Hermitian conjugate, I would suggest using another symbol for it.

Line 18 of the supplementary: (disregard if the paper is modified following the suggestion in comment 1)

The outcome of the singular value decomposition should be the closest orthogonal matrix, and V' is not defined but it should be defined as the transpose of V .

Line 77 of the supplementary:

The sentence "Note that while the variance of the applied perturbation was constant, the observed variance in the perturbed span due to the nonlinear nature of the measurement" appears to be incomplete.

Reviewer #1:

I am very excited for this work, and I genuinely believe this technology and algorithm proposed will be crucial in leveraging long range telecommunications fibers for seismology. The results are timely, and although the concept of leveraging loop-back points builds off another cited recent work (Marra, Science 2022), application here to state-of-polarization sensors will be crucial moving forwards. The possibility of such instruments being minimally intrusive and affordable means this may be the path forward for large scale application.

The article is generally well written and I see very few problems in the way of grammar or explanations. Importantly, however, I feel strongly the authors need to be careful in their wording about having located an earthquake: identifying a fiber span where the signals are strongest is absolutely not equivalent to "precise localization." It is a great leap forward, granted, but for any seismologist reader this is vastly different from a useful earthquake location; even the title of the article could be seen as questionable in this regard.

Other specific questions:

(These are points I offer to potentially improve the manuscript; none do I feel as strongly about and should not impede publication)

Relating to my point about claiming to have located the earthquake, I appreciate the authors' transparency and honesty at line 118, about how some closer spans failed to see the signals. Also, it is clear from the map view (Fig 3a) that Span 41 is not the closest. Nevertheless, this is further evidence the authors should be careful in claims and wording.

Thank you for the positive general comments and for the valuable input regarding clarity. Indeed, we have not demonstrated localization of an earthquake, and while the technique may in principle be able to achieve this by identifying the arrival times of the seismic waves at several spans along the cable (as shown by Marra et al. 2022), we did not demonstrate this, nor did we intend to suggest that we had.

Concerning this, we have noticed the following:

- In the discussion section, we mistakenly used 'seismic event' instead of 'seismic wave'. We have, therefore corrected the following passage:

"We successfully accomplished the precise localization of a seismic event, identifying its location to a single span of fiber between two optical repeaters." to "We successfully

accomplished the localization of a seismic wave, identifying its location to a single span of fiber between two optical repeaters."

- We also revised the following sentence for clarity:
"The ability to localize the seismic wave to within a span enables the observation of the seismic wave move-out (as demonstrated in figure 3e), and may lead to further benefits, such as the determination of the epicenter of a seismic event using a single fiber, and reduced influence of environmental noise compared to cumulative approaches."

We aim to eliminate any potential confusion by the reader with these changes.

Regarding the title, we believe that we have been careful enough by mentioning the ability to localize seismic waves but not the earthquake itself. The localization of the seismic wave within the fiber cable is one of the central claims of the paper and one of the main advantages of the technique, so we feel strongly that it should remain in the title.

The authors list some possible reasons for why other spans may not see the earthquake, additionally potentially there could be something relating to fiber curvature or geometry as mentioned in Fichtner et al (2022, "Introduction to phase transmission..."). Notably, that straight segments should see nothing, though I admit I don't know if this is applicable also for SOP-based measurements, nor how loop-back repeaters would affect this claim.

Thank you for bringing the work by Fichtner and colleagues to our attention, which was missing in our initial bibliography and provides valuable insights towards interpreting effects that may affect local sensitivity of fiber spans to the earthquake wave.

We must note, however, that Fichtner's work is written with phase-measurements in mind, which predominantly measure the effective elongation of the cable. As such, these results do not directly translate to polarization or birefringence-based techniques (as indicated in your question).

These following reasons may play a part on why some spans are unable to detect the seismic wave:

1. The layout and geometry of cable at those spans (e.g. strain coupling and the geometry of the fiber). Regarding the geometry of the fiber layout, as studied by Fichtner et al, note that the treatment for a birefringence-based measurement (such as the eigenvalue method) would have to be different than the simpler, optical path length case explored in their article. First-order effects like elongation, as considered by Fichtner, have a minimal impact on birefringence.
2. Intrinsic features of polarization-based approaches, or the eigenvalue method:
 - a. Changes to the birefringence may depend on the previously existing intrinsic birefringence of the cable. As such, depending on the orientation of the

fast and slow axis of the fiber relative to the fiber displacement, the birefringence may be locally strengthened or attenuated.

b. In addition to the previous point, the eigenvalue method returns only incomplete information about the local changes to the birefringence of the cable: it reports on the change in effective magnitude of the birefringence (eigenvalue), but misses changes to the orientation of the birefringence vector (eigenvector), which, however still affects the output state of polarization.

c. Unlike a DAS system, the forward propagating and reflected light travel through two distinct fibers in the HLLB configuration. Each of the fibers has different intrinsic birefringence magnitudes, different orientation of the birefringence vector, and may experience the earthquake signal differently. Though unlikely, it is possible for the birefringence in the return path to cancel the effects of the forward-path birefringence.

Due to these factors, each span may have a different sensitivity to seismic wave (which, for long-term deployments, may be possible to calibrate)

Note that point 2b is exclusive to the eigenvalue method and distinguishes it from standard SOP measurements. However, in the particular instance of our work, we do not think is the dominant reason for the blind-spots: “Notably, both approaches [SOP and Eigenvalue Method] failed to detect the earthquake at spans closer to the epicenter than the 41st. This suggests that the dominant contribution to the detection limit in any SOP-based method could be due to the complex sensitivity of the local birefringence to different environmental stimuli (e.g., bends, twists, or other effects), the non-linear nature of SOP-based measurements, or variations in mechanical coupling along the cable.”.

We've updated the text to include the role of fiber geometry and cited Fichtner's work for comprehensiveness.

“Notably, both approaches failed to detect the earthquake at spans closer to the epicenter than the 41st. This suggests that the dominant contribution to the detection limit in any SOP-based method could be due to the complex sensitivity of the local birefringence to different environmental stimuli (e.g., bends, twists, or other effects), the non-linear nature of SOP-based measurements, variations in mechanical coupling along the cable, or the geometry/layout of the cable with respect to the induced deformation by the seismic wave [Fichtener et al].”.

Further study is needed to understand the underlying cause of the sensitivity variations and blind spots. For example, a long-term study using simultaneously a phase-based technique (such as the one in Marra et al., 2022) and polarization-based approaches for comparison.

Line 42: Grammar: “On the other hand, polarization-based methods with less stringent hardware requirements have previously been unable of single-span localization” -> unable to [leverage / achieve / take advantage of] single-span approaches?

We have rephrased to hopefully clarify this sentence.

“On the other hand, polarization-based methods with less stringent hardware requirements have previously been unable of single-span localization, limiting their application to either full-span approaches or rudimentary localization techniques limited to a single dominant perturbation in the cable.”

To

“On the other hand, polarization-based methods benefit from less stringent hardware requirements but have thus far been unable to localize the seismic wave to a single-span. The non-commutative nature of birefringence operations has limited these methods to full-cable measurements or, at most, to the localization of a single dominant perturbation occurring along the cable.”

Line 42: Conceptual: You claim that polarization-based methods have been unusable for single-span localization until now, thus motivating the need for the eigenvalue method you propose. But Fig 3C-right, if I understand correctly, is using more traditional direct SOP methods, so what is the limitation there? I guess you cannot isolate Span 41 from that figure? Otherwise what is needed or gained by the eigenvalue approach?

Some limited localization can be achieved with SOP-based methods, as shown in figure 3c (right) – The first span that senses the earthquake can be determined, but all spans following it will be affected. The SOP measurement is, however, fundamentally cumulative. Any environmental perturbation at position M acting on a cable comprised of N spans, will appear on all spans from M to N.

Actual localization of the seismic wave enables (for example) the measurements in sub-figure e (where we see the move-out of the seismic wave to a neighboring span).

As for what is needed by the eigenvalue approach: It requires a set of three measurements at distinct input polarizations, while the SOP method can be done with a single laser shot.

Figure 2: Comment: Some of the symbols and acronyms in the figure are unfamiliar to me as a seismologist. For example, I don't understand what is going on inside the HLLB (loopback inset). This is OK and potentially normally readers of Communication Engineering will be more used to the symbols and acronyms, I only mention it for perspective.

We added a legend to the figure, specifying what are Erbium Doped Fiber Amplifiers and Fiber Bragg Gratings, so that unfamiliar readers can more easily research these.

Line 85: When describing the instrumentation: are signals at each HLLB reflector sent back along the same single fiber, or does each get a unique fiber? I guess it's the former, but as someone less familiar with the technologies involved I can't see how you separate the different HLLB paths.

The HLLB reflects a tiny portion of the launched (forward-propagating) optical wave back through a second fiber. All repeaters have a HLLB which routes light through this second fiber, and discrimination of signals from each repeater is done by time-domain reflectometry.

Assuming that each HLLB is spaced ~ 100 km (200 km roundtrip), and the phase velocity of light in an optical fiber is 2×10^8 m/s, we can expect each reflection to be separated by about 1 ms. We added a few sentences to the methods section, under "Choice of pulse width and repetition rate":

"The pulse width and repetition rate were selected to accommodate the length and repeater spacing of the FUT. The signals from all repeaters (HLLB) paths are transmitted through the same fiber and are discriminated by their time-of-arrival since the respective input pulse launch (see figure 1b). In order to ensure that all reflections arrive before launching the next pulse into the FUT, the repetition rate must be selected as:"

Fig 3e (and other analyses): The frequency range shown to have high energy at 0.25 to 0.35 Hz is rather narrower than I would expect for a M6.0 earthquake. Are there limitations on the instrument noise that limit these observations, or maybe some aspect of the eigenvalue approach? A spectrum or seismogram from a nearby land station (or one of comparable epicentral distance) would help convince me of the measurement capability.

The narrow bandwidth is indeed puzzling and demands further study. In a previous work by some of the authors, strong energy was also observed within a narrow frequency band, with a full-span polarization sensing method (see Fig.3E from Zhan et al, Science, 2021). It may originate, for example, from the generation of a coupled wave which strongly modulates the fiber birefringence. Citing the work by Zhan et al:

"Somewhat unexpectedly, 350 s after the earthquake origin time, another package of strong but lower-frequency (0.3 to 0.8 Hz) waves arrived at the Curie cable (Fig. 3, E and F). Given the waves' slow average speed (~ 2 km/s) and the non-excitation of short-period surface waves from the earthquake at 97 km depth (see fig. S4 for an example of surface waves on SOP), we believe that these late waves are either ocean acoustic waves or Scholte waves converted from

the direct P and S waves near bathymetric features (e.g., slopes, trench) (22, 23) and subseafloor heterogeneities (e.g., fault zones) (7).”

It is possible that the narrow frequency band (0.25-0.35Hz) observed here is also where strong coupled waves are excited. Reducing the noise overall may help reveal the weaker energy outside the band. We will explore this in a future study.

Line 123: Clarification: “The eigenvalue method’s insensitivity to changes...” -> does this imply the eigenvalue method would further outperform SOP approaches, or the other way around? What is meant by “specific stimuli”? If the eigenvalue approach is less sensitive but the direct SOP is more noisy, what will win? I sincerely appreciate this type of analysis and discussion, just I want to make sure I (and readers) understand the implications.

Related to the comment above: It would be great if the authors can comment on the sensitivity of such measurements as relating to earthquakes. Detecting a M6.0 is great for proof-of-concept, but such undersea fibers will only really add seismological value if they can detect things below the range of current traditional instruments (e.g. M1, M2, M3, etc. offshore). I realize this may be beyond the scope of this initial study, so this is not required, I only mention it would be interesting to comment on.

In that specific case, that sentence means to account for the possibility of SOP measurements outperforming the eigenvalue method (in detection limit, not in localization) in some situations.

Since the eigenvalue method only measures changes to the eigenvalues (birefringence strength) but not to the eigenvector orientation (birefringence vector), one can conceive of a case in which a span undergoes a series of deformations along its length, with the net result of not changing the birefringence strength, but changing the orientation of the birefringence vector. In such a case, the eigenvalue method would not be sensitive to any perturbation, but direct measurements of the SOP would.

In a more realistic scenario, any perturbation to the cable will affect both birefringence strength and the orientation of the birefringence vector. SOP changes as a result of both contributions, while the eigenvalue method only measures changes to the strength.

We want to be clear about possible limitations of the eigenvalue technique, and clarify that a comparison in detection limit between the two approaches may be case dependent. In our work, and as we describe in the text, we did not see a relevant difference in SNR from both techniques in our measurements at span 41 to draw any conclusions. For clarification, we added an example to “specific stimuli”, in order to hopefully make that passage more clear.

“Regarding the detection limit of the two tested approaches, we observed no significant SNR differences between the eigenvalue method and direct SOP measurements. Notably, both

approaches failed to detect the earthquake at spans closer to the epicenter than the 41st. This suggests that the dominant contribution to the detection limit in any SOP-based method could be due to the complex sensitivity of the local birefringence to different environmental stimuli (e.g., bends, twists, or other effects), the non-linear nature of SOP-based measurements, or variations in mechanical coupling along the cable, or the geometry/layout of the cable with respect to the induced deformation by the seismic wave [Fichtener].

Nonetheless, it is not easy to draw a direct comparison between the detection limit of both approaches, given the fundamental differences between the eigenvalue and direct-SOP methods: on the one hand, the detection limit when using direct-SOP methods with HLLB will likely be in part determined by the accumulated length of cable up to the interrogated span and the environmental noise acting on the cable (due to the cumulative nature of environmental noise). On the other hand, the eigenvalue method's insensitivity to changes to the birefringence vector orientation suggests potentially lower sensitivity in some scenarios, where the net effect along the span predominantly rotates the birefringence vector, without a great net effect on birefringence strength."

Regarding the second comment, unfortunately our time with the fiber while implementing the eigenvalue method was relatively short, and as a consequence did not store any data from smaller earthquakes. With the current performance of our proof-of-principle demonstrations, it is unlikely that very low magnitude earthquakes (like M1, M2 or M3) at a similar distance from the cable would be detectable, however.

Line 166: Again as one less familiar with the technologies, I don't understand why 9 SOP time-series are generated and why averaging them for Fig 3C-right is appropriate. But possibly this is given in background / cited literature?

There is nothing fundamental about combining the 9 SOP signals. In fact, there seems to be some misunderstanding by the reviewer, as the 9 SOP time-series themselves are not averaged (and that would be wrong). Each is obtained and processed independently to find the signal power in the earthquake band, over time, for each span (i.e., the moving variance of the signal obtained from each repeater). The resulting 2D plots are then averaged. The processing is detailed in the supplementary and methods section.

"The resulting nine time-series are processed independently.

The 2D plot displayed in Fig. 3c (right) is an average of the nine 2D plots obtained from processing each time series."

On the origin of the nine time series: in a normal implementation of the direct SOP approach there wouldn't be 9 time series, but 3. The 3 time series correspond to the 3 polarization components in the normalized Stokes representation, (polarization of light can be represented as 3-dimensional vector, assuming a perfectly polarized wave). As such, launching one state

into the fiber results in 3 time series, one for each of the components of the output Stokes vector.

In our implementation, however, the input is changing between 3 states. So, for each of the 3 possible input polarizations, 3 outputs are generated (one for each of the 3 output Stokes states). This results in nine time series.

In the case of the eigenvalue method, there is just one time series.

Reviewer #2

The paper presents an interesting, telecom-compatible method for localizing geophysical disturbances across a potentially trans-continental fiber cable. The method is an adaptation of what had been already demonstrated in Ref 13 (using the loop-back channel in amplified submarine links to localize disturbances) except that in this case, the measurement is done using polarization changes and conventional telecom lasers (i.e. no need of high-coherence lasers as in Marra's paper).

This implies some advantages as telecom lasers themselves can be used (however, the use of dedicated polarization synthesizers and polarimeters in the measurement channel is needed, which means some unusual hardware in these nodes).

In essence the authors use long pulses with a selectable polarization state (pulses are the size of the span length) and the reflection from each loop-back channel is analyzed as a function of the time of flight of these long pulses. I think the paper is interesting and has to be published. I have several concerns and questions that the authors can surely address in a relatively easy way:

- I had to read too much through the methods section to actually understand the measurement procedure, I am not sure if some of this information could be shifted to the Results section considering that people read Results before Methods. In this case, for people with some skill in optical measurements, the information in Methods is of key importance to understand the process. I suggest to move some of the hardware operation to Results and possibly move part of the matrix treatment to the Methods section.

We appreciate and thank the reviewer for the positive comments. The reviewer raises a good point regarding the organization of the paper.

We moved the post-processing of the eigenvalue method to the results section, after the theoretical section on how to perform single-span localization. We think that this aids in comprehension of the technical part with minimal alteration to the structure of the paper. We also changed the sub-title of that section to "Measurement and Post-processing"

- When it comes to localization, obviously the golden standard in all these systems is using a DAS. Overall, the pulses used here are 300 microseconds long, which is comparatively very long for a DAS (3 orders of magnitude larger). I wonder if a DAS-like architecture with such a relatively long pulse could also give a measurable signal. DAS would have the advantage of being more quantitative and linear than this scheme. I think that an evaluation of the

backscattered energy in such a case could help the authors decide if a poor resolution DAS could also do the same measurement (of course with a more expensive laser).

This is an interesting idea that we have been discussing internally, but it is very different from our current approach, and has several different considerations.

As the reviewer points out, it is true that signals are much longer (higher energy) which could potentially compensate for the added losses from the HLLB (~20 dB). In principle, one could further conceive of performing techniques such as digital pulse compression to further increase the SNR and circumvent these issues.

Compared to a DAS system, however, the peak power of the signals at the input and after each amplification stage is lower than what is often used for DAS (usually close to the modulation instability threshold of 23 dBm/200 mW). Coexistence of high peak-power DAS pulses and telecom channels on the same fiber raises concerns of higher bit-error rates due to cross-phase modulation.

Furthermore, depending on the DAS architecture used, the non-linear phase contribution of the reflector regions to the strain signal depends on the input pulse width. The pulse width and gauge length need to be carefully considered so that the non-linear phase contribution (originating from displacement in scattering centers within the pulse regions) does not dominate the linear contribution due to the phase evolution over the gauge length, between the two pulse positions (start and end of each gauge length). Pulse widths are typically selected to be shorter than the gauge length for this reason, and it is unclear if such long pulses may lead to high non-linearity in phase measurements. This is also addressable, however, as there have been a few recent works on using multi-frequency measurements to mitigate the nonlinear contribution (Ogden, et al, 2021, Scientific Reports).

One final concern is the increased noise floor due to the constant stream of ASE light being backscattered and transmitted by the HLLB path (from the whole fiber). This may further increase the power demands for retrieving a measurable signal. Also, each span is roughly of the length of the full range capable of being interrogated by current DAS systems. It is conceivable that, even in an optimistic scenario, there are some blind spots near the end of each span.

Finally, as the reviewer pointed out, this would imply using a high coherence laser – coherence length of at least the spatial resolution. In this work, we were aiming for an implementation that was compatible with telecom lasers and not reliant on coherent detection schemes.

Again, this is an interesting idea which could possibly lead to a publication at a later point, but it is far beyond the current work, and would require a significant research effort.

- I am sure that during the measurement campaign there were other disturbances of smaller magnitude that could be recorded along the used cable (this is a seismically very active region). Please provide information of what is the minimum magnitude of event that could be detected

in the measurement campaign done here. Showing the magnitude 6 event is interesting, but giving the actual sensitivity threshold would be necessary to comparatively assess this method and the others published in the literature.

This comment echoes one of the comments made by Reviewer 1. Unfortunately, our time with the fiber since having the eigenvalue method implemented correctly was limited, and we did not store data from other events - We only retrieved data from the system when we saw a relatively strong earthquake happening, as a means of validating the technique.

For this proof of concept, it seems unlikely that this method would be sensitive to lower-magnitude earthquakes (M1 to M4) occurring at similar distances, judging from the magnitude of the received signals against the observed noise. Perhaps with further optimization or additional processing this could be achieved, but goes beyond the scope of this proof of concept.

- Sampling is very low (sub Hz in this case, potentially 2-3 Hz if the hardware had no delay times) as the reflections from all the repeater spans have to be collected and 3 polarization states have to be swept. Please comment if there is any room for increasing the sampling while keeping the same constraints in terms of fiber size.

This is a limitation of the eigenvalue method (as of DAS system or other roundtrip time-of-flight techniques). In this case, there is an additional penalty as the sampling will be 3 times slower than single-shot techniques.

Fundamentally, the measurement rate is limited by the fiber length, as pointed out by the reviewer. Increasing the complexity of the hardware may enable some mitigation of this limitation (as can be for other roundtrip techniques, using multifrequency probing or coding, for example), but that would require further research.

- Of course the interest of gathering measurements across many points is using array methods. However, considering the "nonlinear" nature of these polarization measurements, would this be compatible with array processing?

Coherent array signal processing techniques, such as beamforming, are not usable unless the nonlinearity can be calibrated or accounted for. Similarly to the previous answer, one possible option is to perform the same measurement at multiple optical frequencies to attempt to overcome the nonlinearity, at the cost of increasing hardware cost and processing complexity.

Nevertheless, there is something to be gained from having localization information, even with incoherent measurements between channels. Namely, the time-of-arrival of the seismic wave at different spans can be measured. We were able to observe the seismic wave move-out through two spans, which may eventually lead to localization of the earthquake origin.

Reviewer #3:

Review of "Localization of Seismic Waves in Submarine Fiber Optics Using Polarization-only Measurements," by Luis Costa et al.

The manuscript presents a report on the detection and localization of an earthquake using an undersea fiber optics infrastructure. It appears to be a valuable addition to the rapidly growing literature on this topic. I am inclined to endorse its publication pending the proper addressing of my concerns outlined below:

1) My primary concern relates to the authors' use of singular value decomposition (SVD) as an intermediate step to obtain a polar decomposition for extracting the unitary part of the transmission matrix. While this procedure is standard in Jones space, it may not be suitable in Stokes space. To illustrate this, consider the simple case of combining a (partial) polarizer represented in Jones space by a (positive definite) matrix A and a concatenation of waveplates represented by an arbitrary unitary matrix U .

In Jones space, the transmission matrix T is given by $T = UA$. The polar decomposition of T is either $T = U A$ (right polar decomposition) or $T = B U$ (left polar decomposition) and is unique. Consequently, U is also unique, and applying SVD would yield the exact result, providing the unitary matrix U and, in Stokes space, the rotation matrix corresponding to U .

However, if the SVD is directly applied in Stokes space, it would not return the 3 by 3 rotation matrix corresponding to U . This limitation arises because representing a pure polarizer as a linear operator is not possible within the 3-dimensional Stokes space. To maintain the linearity of the representation, it becomes necessary to extend the Stokes space with an extra dimension representing the total power and replace the matrices that represent rotations in Stokes space with 4 by 4 Mueller matrices. In the extended space, the unitary component of the decomposition is the direct sum of a rotation in the 3-dimensional Stokes space and an identity in the fourth coordinate. This makes not straightforward the application of the SVD to extract from the transmission matrix the unitary part of the concatenation.

Of course, the use of the SVD in the 3-dimensional Stokes space would still produce, for the concatenation polarizer-waveplates, a unitary matrix, but in most cases this unitary matrix includes the polarization rotation induced by the partial polarizer, which would instead be filtered out if the SVD is applied in Jones space.

Earthquakes primarily affect fiber propagation by inducing changes in the fiber's refractive index and birefringence, thereby impacting the unitary part of the transmission matrix. On the other hand, polarization-dependent loss is mainly caused by lumped devices and remains substantially time-independent. By applying the SVD in Stokes space, crosstalk is generated

between the time-independent polarization-dependent loss and the time-dependent unitary part of polarization rotation. Consequently, this crosstalk has the potential to significantly reduce the sensitivity to time-dependent birefringence changes.

Given that it is not challenging to extract the transmission matrix in Jones space from the data, the authors should reconsider their data processing approach and extract the unitary part of the fiber propagation by applying the SVD in Jones space rather than in Stokes space.

Thank you for the careful analysis and suggestion.

The reviewer raises a great point concerning the processing of the data in Stokes space, which we had not taken into consideration. In our manuscript, we treated the normalization of the Stokes vectors and the SVD as simple processing steps to obtain unitary (orthogonal) matrices from noisy estimations and partially polarized light.

As the reviewer correctly points out, there are physical implications to this kind of processing, due to the presence of polarization dependent loss on the cable. By not working with the full Muller matrices, and instead a partial representation of transmission matrices in Stokes space, we include polarization changes from both rotation of the polarization state and from polarization dependent loss. This may contribute as a loss of sensitivity on our measurements

We appreciate the reviewer's suggestion to perform the SVD in Jones space or attempt to work with the full 4x4 Muller matrix. However, as we have used a Stokes receiver in our experiments, we do not have access to the Jones data. Additionally, we are unable to recover the 4D Muller matrices from our measurements since we are building each matrix out of only 3 acquisitions (or a 3D basis). As such, it is not immediately clear to us how we could perform that processing from the current data.

Nevertheless, we do agree that the reviewer is fundamentally correct in his assessment, and this must be addressed in the manuscript. We added the following paragraph to the discussion section of the text.

Additionally, the sensitivity of the eigenvalue technique may potentially be improved by using a Jones receiver or by recovering the full Muller matrix. Currently, by acquiring three sets of Stokes components, normalizing each of them, and calculating the closest unitary matrix, we are making an assumption of no polarization dependent loss in each span. The polarization rotation originating from environmental changes (i.e., changes to the birefringence) will be combined with the apparent rotation originating from polarization-dependent loss of the lumped elements as the same signal (which is largely time-independent, and not directly correlated to environmental changes).

We would like to highlight that this does not invalidate our method and results, but does bring forward an underlying assumption to our processing that we had not previously mentioned in the manuscript, and points a clear path towards optimization and further work which we had not previously considered.

2) The experiment's specific details regarding the system where it was performed have not been provided in the report. However, it appears that the system under test bears a striking resemblance to Curie, the system described in [14]. The only discernible difference is the location of one of the system's terminals, with one being in Santiago instead of Valparaiso. To ensure transparency and enable readers to thoroughly understand the characteristics of the system under test, it is crucial to provide this information. Additionally, the report should explicitly state whether the data were collected from the Santiago or Los Angeles terminal.

We thank the reviewer for the careful read of the paper. This is in fact, a mistake (which we have now corrected). Indeed, one of the system's terminals is in Valparaiso.

We also added the information on the name of the cable, and on which of the terminals the interrogation setup is located at.

In the results section:

“On December 11, 2022, at 14:31:29 UTC, a magnitude 6.0 earthquake occurred in Guerrero, Mexico, which we captured on the Curie transoceanic fiber cable, which connects Los Angeles (California) to Valparaiso (Chile). The interrogation setup (situated in the Los Angeles terminal) is depicted in Fig.2 , and includes a telecommunication transponder used to send linearly polarized optical pulses through a polarization synthesizer on the emitter side, and a polarimeter on the receiver side, which is used to evaluate the state of polarization of the received reflections.”

Minor comments:

Line 70: (disregard if the paper is modified following the suggestion in comment 1). The authors' analysis is conducted in Stokes space, not in Jones space. Consequently, U represents an arbitrary matrix with real entries describing a (proper) rotation in Stokes space, which is a special case of an orthogonal matrix, not an arbitrary (complex) unitary matrix. This distinction is important as it ensures that readers are given the immediate perception that the analysis takes place in Stokes space, not in Jones space.

This is a good point, for the reasons described of avoiding confusion with a Jones space implementation.

We changed the following sentence: where s is the normalized Stokes vector representing the SOP of the input pulse, and A_m is the real-valued rotation (orthogonal) matrix that describes the cumulative birefringence effects of the complete round-trip to and from the m -th repeater (Fig. 1a).

In the following sentence, however, we kept the word unitary, because that is a general result. The matrix can be any unitary, but in this case will always be a rotation matrix.

“where $U = A^{\text{fwd}}_{(m-1)}$ can be any (unknown) unitary matrix.”

Line 112:

Would be beneficial that the definition of crosstalk is given the first time it is introduced and discussed. The reader is not exposed to the mathematical definition of crosstalk until hitting the figure caption of Fig. 4 of the supplementary material.

We changed the first mention of crosstalk in the main text to the following.

“However, while the earthquake signal is visible in every span following the 41st when using the direct SOP approach, our eigenvalue method localizes the measurement to a single span with minimal crosstalk (defined as the increase in signal noise power in the earthquake frequency band to subsequent fiber locations). We observe a median value of ~1 dB of crosstalk, Fig. 3d.”

Line 157: (disregard if the paper is modified following the suggestion in comment 1)

Again, the U and V matrix are real, so that V^* should be the transpose of V . Since the star is usually reserved for Hermitian conjugate, I would suggest using another symbol for it.

Changed to V^T , as per the reviewer's suggestion.

Line 18 of the supplementary: (disregard if the paper is modified following the suggestion in comment 1)

The outcome of the singular value decomposition should be the closest orthogonal matrix, and V' is not defined but it should be defined as the transpose of V .

Changed to V^T , in accordance to the main text.

Line 77 of the supplementary:

The sentence “Note that while the variance of the applied perturbation was constant, the observed variance in the perturbed span due to the nonlinear nature of the measurement” appears to be incomplete.

We rewrote that paragraph, in hopes of making it clearer:

We define crosstalk as the median variance of the signal observed in all (unperturbed) spans located after the perturbed span, normalized to the variance of the signal observed in the perturbed span (which may change between runs of the simulation, due to the nonlinearity of eigenvalue measurements). In figure S4a, we plot the crosstalk against the orthogonality figure

of merit (Q) and the maximum birefringence change between consecutive acquisitions (as a measure of non-stationarity).

REVIEWERS' COMMENTS:

Reviewer #1 (Remarks to the Author):

The authors gave a very thorough response to my previous comments and I am satisfied with the clarifications they have made. Especially regarding the wording about having located seismic waves instead of the event - I think the claims are now well matched by the work presented.

I was unable to assess some of the technical details initially, but it seems the other reviewers were able to prod quite deeply about the applicability, for example as in reviewer 3's questions about SVDs in Stokes and Jones spaces.

In any case, I think the paper reads nicely and is in a good form for publication.

Reviewer #2 (Remarks to the Author):

I am overall satisfied with the current version of the work. I agree with one of the other reviewers that the narrowband response of the system should be evaluated more in depth (probably in future work) and may point to some limitations imposed by the mechanical coupling of the seismic waves to the cable. Nevertheless, the work is very interesting and may be useful in other scenarios.

Reviewer #3 (Remarks to the Author):

The authors have adequately addressed my comments, and I fully support the publication of the paper.